# Primary care utilization among telehealth users and non-users at a large urban public healthcare system

**Kevin Chen** [1,2]*, **Christine Zhang**[1], **Alexandra Gurley**[1], **Shashi Akkem**[1], **Hannah Jackson**[1,2]

1 New York City Health + Hospitals, New York, New York, United States of America, 2 Division of General Internal Medicine and Clinical Innovation, New York University Grossman School of Medicine, New York, New York, United States of America

* research@kchenmd.com

**Data Availability Statement:** Data cannot be shared publicly because they contain potentially identifying and sensitive patient information. Data are available from the New York City Health +

## Abstract

Telehealth services may improve access to care, but there are concerns around whether availability of telehealth may increase care utilization. We assessed whether usage of telehealth was associated with differential primary care utilization at a large, urban public healthcare system. Using electronic health record data from 23 primary care clinics, we categorized patients as telehealth users and non-users. Then, we compared the number of visits per patient between groups using Welch's t-tests while stratifying by comorbidity count. We used multivariable Poisson regression to test for associations between telehealth usage and visit count while controlling for other demographic factors. Compared with telehealth non-users, telehealth users had approximately 1 more primary care visit per patient over the year regardless of comorbidity count or other patient characteristics. Availability of telehealth services may be associated with increased primary care utilization in a safety-net setting, though further research on outcomes, costs of care, and patient and clinician experiences is needed to better inform decisions regarding provision and reimbursement of telehealth services.

## Introduction

Telehealth visits (specifically synchronous audio-only and audio-video episodes of care between patients and clinicians) may improve access to care by removing certain barriers such as time and cost related to transportation or logistics for childcare or work coverage [1]. In a primary care setting, these visits can be used to deliver medical evaluation and management for acute, chronic, and preventive care needs—analogous to traditional scheduled in-person visits for initial or follow-up evaluation [2]. However, health systems and payors may hesitate to provide or cover telehealth at the same rate as in-person visits in part due to concerns around potential to increase overall healthcare utilization [3, 4]. Whether telehealth visits can functionally substitute for traditionally in-person visits or whether they, for better or worse, create additional encounters, is unclear. During the coronavirus disease (COVID) pandemic,

Hospitals Institutional Data Access Committee (contact via AmbCareResearch@nychhc.org; star.nychhc.org) for researchers who meet the criteria for access to confidential data.

**Funding:** KC and HJ received a grant (21-12977) from the New York State Health Foundation (https://nyshealthfoundation.org/). The funders had no role in study design, data collection and analysis, decision to publish, or preparation of the manuscript.

**Competing interests:** I have read the journal's policy and the authors of this manuscript have the following competing interests: KC and HJ received a grant (21-12977) from the New York State Health Foundation. KC, CZ, AG, SA, and HJ are employed by New York City Health + Hospitals. The authors have no other competing interests to declare. New York City Health + Hospitals provided support in the form of salaries for authors KC, CZ, AG, SA, and HJ, but did not have any additional role in the study design, data collection and analysis, decision to publish, or preparation of the manuscript. The specific roles of these authors are articulated in the 'author contributions' section. This does not alter our adherence to PLOS ONE policies on sharing data and materials.

many regulatory restrictions on telehealth in the United States were paused, allowing more widespread usage of telehealth [3].

In this setting, we sought to assess whether access to telehealth affected primary care utilization at a large, urban public healthcare system.

## Methods

We conducted an observational study of electronic health record data for patients with scheduled primary care visits from July 1, 2020 to June 30, 2021 at 23 adult primary care clinics of New York City Health and Hospitals. This period represents when local COVID cases were past initial peak and telehealth visits were available to patients electively instead of preferentially. All adult primary care patients were eligible for participation in telehealth, and we included all adult primary care patients who had primary care encounters during the study interval in our analysis. There were no set exclusion criteria for telehealth participation. Clinicians and patients used their discretion when scheduling primary care appointments to determine whether a telehealth visit was appropriate or preferable for a given concern.

We classified patients as telehealth users and non-users. Telehealth users conducted ≥1 audio-only or video visit in primary care during the sample period. Telehealth non-users exclusively had in-person encounters.

The primary outcome was the average number of completed primary care visits per patient over the one-year timeframe.

We collected patient characteristics (age, sex, race/ethnicity, language, insurance, and number of Elixhauser comorbidities [5]) and compared them between groups using chi-squared tests.

We hypothesized that comorbidity count would be most associated with number of visits, so we stratified patients by quintiles of comorbidity count and compared the average number of completed primary care visits per patient between telehealth users and non-users using two-sided Welch's t-tests.

Finally, we used multivariable Poisson regression to estimate differences in the primary outcome by telehealth user status while controlling for all measured patient characteristics. Regression assumptions were confirmed, and model fit was tested using a chi-squared goodness-of-fit test (p = 1.00).

We used Stata SE, version 15 (StataCorp), for all analyses. Our threshold for statistical significance was $p < 0.05$. This study was exempt from full review by the Biomedical Research Alliance of New York institutional review board with a waiver of informed consent to access anonymized medical record data.

## Results

There were 569,724 visits by 225,147 patients over the one-year study period across all 23 clinic sites. Of these patients, 133,830 (59.4%) were telehealth users. Compared to telehealth non-users, telehealth users were more likely to be older, female, Asian, Medicare beneficiaries, and have more comorbidities; they were less likely to be Black, commercially insured, or uninsured (Table 1; $p < 0.001$).

The average (standard deviation) number of primary care visits were 2.9 (1.7) for telehealth users and 1.9 (1.3) for non-users. Compared to telehealth non-users, telehealth users had 1 more primary care visit per patient per year regardless of comorbidity count (Table 2; $p < 0.001$). Among telehealth users, the average proportion of visits that were conducted via telehealth was 0.68 (0.28).

**Table 1. Patient characteristics.**

| Demographic, N (%) | Telehealth Users N = 133,830 | Telehealth Non-users N = 91,317 | p |
|---|---|---|---|
| Age, years | | | <0.001 |
| 18–44 | 36,374 (27.2) | 32,584 (35.7) | |
| 45–64 | 60,647 (45.3) | 39,036 (42.8) | |
| ≥65 | 36,809 (27.5) | 19,697 (21.6) | |
| Female | 82,997 (62.0) | 52,356 (57.3) | <0.001 |
| Race/Ethnicity | | | <0.001 |
| White | 11,645 (8.7) | 8,204 (9.0) | |
| Black | 43,134 (32.2) | 33,621 (36.8) | |
| Hispanic | 46,877 (35.0) | 31,034 (34.0) | |
| Asian | 11,366 (8.5) | 4,736 (5.2) | |
| Other | 20,808 (15.6) | 13,722 (15.0) | |
| Primary Language | | | <0.001 |
| English | 75,231 (56.2) | 52,744 (57.8) | |
| Spanish | 47,986 (35.9) | 31,883 (34.9) | |
| Other | 10,613 (7.9) | 6,690 (7.3) | |
| Insurance | | | <0.001 |
| Commercial | 20,931 (15.6) | 16,394 (18.0) | |
| Medicaid | 63,962 (47.8) | 43,270 (47.4) | |
| Medicare | 27,943 (20.9) | 14,517 (15.9) | |
| Other | 990 (0.7) | 551 (0.6) | |
| Uninsured | 20,004 (15.0) | 16,585 (18.2) | |
| Elixhauser Comorbidity Count | | | <0.001 |
| 0 | 26,858 (20.1) | 25,705 (28.2) | |
| 1 | 36,300 (27.1) | 26,912 (29.5) | |
| 2 | 33,446 (25.0) | 20,050 (22.0) | |
| 3 | 20,515 (15.3) | 10,658 (11.7) | |
| ≥4 | 16,711 (12.5) | 7,992 (8.8) | |

While controlling for patient characteristics, the incidence rate ratio for number of primary care visits between telehealth users and non-users was 1.44 [95% confidence interval 1.43, 1.45]. In other words, after accounting for demographic differences, telehealth users had 0.88 [0.87, 0.89] more primary care visits over the year than telehealth non-users (S1 Table).

## Discussion

In this study of primary care utilization at a large urban public healthcare system, we found that telehealth users had, on average, approximately 1 more visit per patient over one year than

**Table 2. Number of primary care encounters by telehealth users and non-users stratified by Elixhauser comorbidity count.**

| Elixhuaser Comorbidity Count | Number of Primary Care Encounters by Telehealth Users, Mean (SD) | Number of Primary Care Encounters by Telehealth Non-users, Mean (SD) | p |
|---|---|---|---|
| 0 | 2.1 (1.3) | 1.4 (0.8) | <0.001 |
| 1 | 2.7 (1.5) | 1.8 (1.0) | <0.001 |
| 2 | 3.1 (1.6) | 2.1 (1.3) | <0.001 |
| 3 | 3.4 (1.7) | 2.5 (1.5) | <0.001 |
| ≥4 | 3.9 (2.2) | 2.8 (2.0) | <0.001 |
| Total | 2.9 (1.7) | 1.9 (1.3) | <0.001 |

telehealth non-users regardless of comorbidity count and demographics. This suggests that availability of telehealth visits in primary care may be associated with increased individual primary care utilization. Our findings contrast with those of prior studies that included primary care during a contemporary period which showed no increase in utilization [6–8], though our study differs in methodology and population demographics.

We uniquely focus on utilization in primary care only whereas prior literature that included primary care [6–8] aggregated outpatient specialties together. The impact of telehealth on different service lines may be different; for example, compared with historical averages, behavioral health services may have experienced rapid adoption and increased total visit volumes with telehealth during the pandemic [6], whereas surgical specialties may have had slower adoption and depressed visit volumes despite telehealth availability during the pandemic [9]. We also chose not to use a historical comparison for visit volume since total visit volumes during the study period were affected by the pandemic. Another difference is in patient demographics; for example, in prior studies that included primary care, 30–77% of patients were White, 8–40% Hispanic, 6–8% Black, and 4–10% Asian [6–8]. In contrast, 9% of our patients were White, 34% Black, 34% Hispanic, and 7% Asian. One study of a large, integrated healthcare system demonstrated that later in the pandemic, while total visit volumes generally returned towards historical trends, not all age, racial, or income groups had similar trajectories [7].

We are unable to comment on whether the observed increase in primary care utilization is due to misusage or reduced efficacy of telehealth services versus more frequent engagement due to improved access to care. We hypothesize that the observed difference may be because reductions in barriers to primary care, such as transportation, may affect groups within safety-net populations disproportionally [10]. Additionally, since primary care is often the first point of contact for ambulatory healthcare, this increased access may have a specialty-specific effect of increasing utilization.

While payors and policymakers may be concerned that telehealth availability may increase healthcare costs via additive service utilization, increased primary care utilization in the context of telehealth availability may not be a negative consequence if patients can have more health concerns addressed in a safe and effective manner. Changes in reimbursement structure in the United States towards more value- or outcomes-based payment may allow health systems and other entities to provide telehealth services while tempering costs for payors regardless of visit count.

Future studies of telehealth and healthcare utilization may consider using prospective or cohort designs to examine associations more robustly. And, further research will be needed on clinical and experiential outcomes of telehealth encounters to inform longer-term policies around provision and reimbursement of telehealth.

## Limitations

Our study examines utilization behavior during a pandemic, which may not be applicable to non-pandemic times. It is possible that some patients who were higher utilizers of the health system elected to change some visits to telehealth due to concern about travelling during a pandemic. We used comorbidity count as a rough proxy for both expected healthcare utilization (more chronic diseases may correlate with more medical service use [11]) and past health system engagement (more encounters with the healthcare system may correlate with more diagnosis codes entered into a patient's chart) and stratified by comorbidity count to account for these factors. However, telehealth users and non-users may differ in propensity to seek care for reasons unaccounted for by their number of comorbidities or the demographics measured in

this study, such as distance to the clinic, education level, or socioeconomic status [12]. Further, we did not account for disease severity, which may also impact the number of encounters. Data were from a single urban healthcare system, though they represent multiple primary care facilities with diverse patients. We used an observational study design, which limits our ability to infer causality.

## Conclusions

Availability of telehealth services may be associated with increased primary care utilization in safety-net clinics. Further research on clinical outcomes, costs of care, patient and clinician experiences is essential to better inform policymakers' and payors' decisions regarding reimbursement of telehealth services.

## Supporting information

**S1 Table. Full regression model for number of primary care visits by telehealth user status.** (DOCX)

## Author Contributions

**Conceptualization:** Kevin Chen, Hannah Jackson.

**Data curation:** Christine Zhang, Alexandra Gurley, Shashi Akkem.

**Formal analysis:** Kevin Chen.

**Funding acquisition:** Kevin Chen, Hannah Jackson.

**Investigation:** Kevin Chen, Christine Zhang.

**Methodology:** Kevin Chen, Hannah Jackson.

**Project administration:** Kevin Chen.

**Supervision:** Kevin Chen, Hannah Jackson.

**Validation:** Kevin Chen, Christine Zhang, Alexandra Gurley, Shashi Akkem.

**Writing – original draft:** Kevin Chen, Hannah Jackson.

**Writing – review & editing:** Kevin Chen, Christine Zhang, Alexandra Gurley, Shashi Akkem, Hannah Jackson.

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
