## [Decision Letter · Decision Letter 0]

20 Apr 2022

PONE-D-22-05119Primary care utilization among telehealth users and non-users at a large urban public healthcare systemPLOS ONE

Dear Dr. Chen,

Thank you for submitting your manuscript to PLOS ONE. After careful consideration, we feel that it has merit but does not fully meet PLOS ONE’s publication criteria as it currently stands. Therefore, we invite you to submit a revised version of the manuscript that addresses the points raised during the review process.

We look forward to receiving your revised manuscript.

Kind regards,

Esther Cubo Delgado

Academic Editor

PLOS ONE

Journal Requirements:

3. Thank you for stating the following in the Competing Interests/Financial Disclosure* (delete as necessary) section: 

( I have read the journal's policy and the authors of this manuscript have the following competing interests: KC and HJ received a grant (21-12977) from the New York State Health Foundation. KC, CZ, AG, SA, and HJ are employed by New York City Health + Hospitals.)   

We note that one or more of the authors are employed by a commercial company: name of commercial company. 

5. Please include your full ethics statement in the ‘Methods’ section of your manuscript file. In your statement, please include the full name of the IRB or ethics committee who approved or waived your study, as well as whether or not you obtained informed written or verbal consent. If consent was waived for your study, please include this information in your statement as well

Additional Editor Comments:

In this article, the authors retrospectively compared the number of telehealth visits vs. in-office visits in a large sample in New York. For international readers, it would be very convenient to define the different types of health care provider systems used in the US and included in this article. As an academic editor I completely agree with the comments provided by the first reviewer. I would discuss the limitations more deeply because it would give an idea how to extrapolate the results to other communities. It is very important to understand how the sample was selected (inclusion and exclusion criteria), and to add other confounding variables if they are known (distance to the health care facility). Other important aspect for publishing a research article is the discussion, comparing the authors´results with the previous literature. Surprisingly, there are very few references.

I would suggest the authors to include all the comments provided in this feed back and to send the article again for a second review.

Reviewers' comments:

Reviewer's Responses to Questions

**Comments to the Author**

1. Is the manuscript technically sound, and do the data support the conclusions?

Reviewer #1: Yes

2. Has the statistical analysis been performed appropriately and rigorously? 

Reviewer #1: Yes

3. Have the authors made all data underlying the findings in their manuscript fully available?

Reviewer #1: No

4. Is the manuscript presented in an intelligible fashion and written in standard English?

Reviewer #1: Yes

5. Review Comments to the Author

Reviewer #1: This paper discussed about the association between the use of telehealth and health care utilization. This topic is important especially during crisis (e.g. COVID). The paper is also very concise. My comments are as follows:

1/ to benefit international readers, the authors may want to describe more how telehealth is used in these clinics. Telemedicine ranged from simply transmission of observations (e.g. blood pressure readings) to full consultations. When the authors said telehealth users, what does this include?

Will primary care clinic visits be advised after a telehealth consultation? That will explain the difference of no of clinic visits between the both groups. (e.g. skin lesions that cannot be seen clearly during the telehealth consultation)

Similarly, will telehealth be more likely advised (by healthcare professionals) when the diseases are poorly controlled? that will explain the difference between the two groups

2/ Any inclusion and exclusion criteria of participants? If the "non-users" include simple patients who have good past health and only consult the clinic for episodic problems (e.g. common cold), of course they will have very few clinic visits.

3/ in the tables, I suggest to replace "something else" by "others"

4/ I think the main weakness is that there are many unmeasured parameters to explain the results. The authors may want to discuss possible important parameters and address/discuss this in more details in the discussion; e.g.

- disease severity: people with poorly controlled diseases may want more contact time with the healthcare system. Therefore, they may have more telehealth and also more consultations. No of co-morbidities is not exactly the same as disease severity

- Education level

- level of social support

5/ As this is a cross-sectional study, the causal relationship cannot be established. it is possible that those patients, who had frequent clinical visits, changed some of the visits to telehealth during COVID pandemic

6/ The authors may also want to elaborate more about clinical/research implications of their results. RCTs and cohorts maybe a better choice to answer whether implementation of telehealth systems would increase healthcare utilization

6. PLOS authors have the option to publish the peer review history of their article (what does this mean?). If published, this will include your full peer review and any attached files.

Reviewer #1: No

---

## [Author Response · Author response to Decision Letter 0]

14 Jun 2022

Additional Editor Comments: 

In this article, the authors retrospectively compared the number of telehealth visits vs. in-office visits in a large sample in New York. 

1. Comment: 

For international readers, it would be very convenient to define the different types of health care provider systems used in the US and included in this article. 

Response:

As suggested, we have defined what we are referring to when discussing telehealth. Please see our response to Reviewer #1, Comment #1 for full details.

Change:

Please see our response to Reviewer #1, Comment #1 for full details.

2. Comment: 

As an academic editor I completely agree with the comments provided by the first reviewer. I would discuss the limitations more deeply because it would give an idea how to extrapolate the results to other communities. 

Response:

We appreciate your and the reviewer’s comments on expanding the Limitations discussion. We have added to the Limitations section in response to Reviewer #1, Comments #4-6.

Change:

Please see our response to Reviewer #1, Comments #4-6 for full details.

3. Comment: 

It is very important to understand how the sample was selected (inclusion and exclusion criteria), and to add other confounding variables if they are known (distance to the health care facility). 

Response:

We have clarified what the inclusion and exclusion criteria were in response to Reviewer #1, Comment #2. Unfortunately, additional confounding variables as suggested in Reviewer #1, Comment #4 were not available, but we did include further discussion of this in the Limitations.

Change:

Please see our response to Reviewer #1, Comments #2 and #4 for full details.

4. Comment: 

Other important aspect for publishing a research article is the discussion, comparing the authors´ results with the previous literature. Surprisingly, there are very few references.

I would suggest the authors to include all the comments provided in this feed back and to send the article again for a second review.

Response:

Thank you for this suggestion. We have added more comparisons to existing literature and expanded our discussion with more information about our hypotheses on interpretations of our findings. We have also added more citations throughout the text.

Change:

In Discussion: 

Our findings contrast with those of prior studies that included primary care during a contemporary period which showed no increase in utilization [6-8], though our study differs in methodology and population demographics. 

We uniquely focus on utilization in primary care only whereas prior literature that included primary care [6-8] aggregated outpatient specialties together. The impact of telehealth on different service lines may be different; for example, compared with historical averages, behavioral health services may have experienced rapid adoption and increased total visit volumes with telehealth during the pandemic [6], whereas surgical specialties may have had slower adoption and depressed visit volumes despite telehealth availability during the pandemic [9]. We also chose not to use a historical comparison for visit volume since total visit volumes during the study period were affected by the pandemic. Another difference is in patient demographics; for example, in prior studies that included primary care, 30-77% of patients were White, 8-40% Hispanic, 6-8% Black, and 4-10% Asian [6-8]. In contrast, 9% of our patients were White, 34% Black, 34% Hispanic, and 7% Asian. One study of a large, integrated healthcare system demonstrated that later in the pandemic, while total visit volumes generally returned towards historical trends, not all age, racial, or income groups had similar trajectories [7].

[…] Additionally, since primary care is often the first point of contact for ambulatory healthcare, this increased access may have a specialty-specific effect of increasing utilization.

In references:

1. Reed ME, Huang J, Parikh R, Millman A, Ballard DW, Barr I, et al. Patient-Provider Video Telemedicine Integrated With Clinical Care: Patient Experiences. Ann Intern Med. 2019 Aug 6;171(3):222-224. 

2. Reed M, Huang J, Graetz I, Muelly E, Millman A, Lee C. Treatment and Follow-up Care Associated With Patient-Scheduled Primary Care Telemedicine and In-Person Visits in a Large Integrated Health System. JAMA Netw Open. 2021 Nov 1;4(11):e2132793.

5. Quan H, Sundararajan V, Halfon P, Fong A, Burnand B, Luthi JC, et al. Coding algorithms for defining comorbidities in ICD-9-CM and ICD-10 administrative data. Med Care. 2005 Nov;43(11):1130-9.

9. Chao GF, Li KY, Zhu Z, McCullough J, Thompson M, Claflin J, et al. Use of Telehealth by Surgical Specialties During the COVID-19 Pandemic. JAMA Surg. 2021 Jul 1;156(7):620-626.

11. Fortuna D, Berti E, Moro ML. Multimorbidity epidemiology and health care utilization through combined healthcare administrative databases. Epidemiol Prev. 2021 Jan-Apr;45(1-2):62-71.

12. Reed ME, Huang J, Graetz I, Lee C, Muelly E, Kennedy C, et al. Patient Characteristics Associated With Choosing a Telemedicine Visit vs Office Visit With the Same Primary Care Clinicians. JAMA Netw Open. 2020 Jun 1;3(6):e205873.

---

Reviewer #1: 

This paper discussed about the association between the use of telehealth and health care utilization. This topic is important especially during crisis (e.g. COVID). The paper is also very concise. My comments are as follows:

1. Comment: 

To benefit international readers, the authors may want to describe more how telehealth is used in these clinics. Telemedicine ranged from simply transmission of observations (e.g. blood pressure readings) to full consultations. When the authors said telehealth users, what does this include?

Will primary care clinic visits be advised after a telehealth consultation? That will explain the difference of no of clinic visits between the both groups. (e.g. skin lesions that cannot be seen clearly during the telehealth consultation)

Similarly, will telehealth be more likely advised (by healthcare professionals) when the diseases are poorly controlled? that will explain the difference between the two groups

Response:

Thank you for this comment; we have added a more specific definition in the introduction of what services we are referring to under “telehealth services”. Briefly, we are referring to synchronous audio-only and audio-video care episodes between patients and clinicians for the purpose of medical evaluation and management of acute, chronic, or preventive care needs. To make it clearer that we are discussing such encounters, we have also exchanged the word “services” for “visits” (i.e. telehealth visits instead of telehealth services) to emphasize the analogy to traditionally in-person primary care visits.

Regarding the reviewer’s question of whether clinic visits will be advised after a telehealth visit, that is one of the key questions we are trying to explore. Thus, we have added a sentence in the introduction addressing this. We caveat that more visits are not necessarily bad—it may be that previously, patients were not seeking care often enough due to general barriers to care. 

Finally, we have added a few references to expand on the theoretical benefit of telehealth visits and a similar use case to ours. 

Change:

In Introduction, paragraph 1:

Telehealth visits (specifically synchronous audio-only and audio-video episodes of care between patients and clinicians) may improve access to care by removing certain barriers such as time and cost related to transportation or logistics for childcare or work coverage [1]. In a primary care setting, these visits can be used to deliver medical evaluation and management for acute, chronic, and preventive care needs—analogous to traditional scheduled in-person visits for initial or follow-up evaluation [2].

…at the same rate as in-person visits in part due to…

In references:

1. Reed ME, Huang J, Parikh R, Millman A, Ballard DW, Barr I, et al. Patient-Provider Video Telemedicine Integrated With Clinical Care: Patient Experiences. Ann Intern Med. 2019 Aug 6;171(3):222-224. 

2. Reed M, Huang J, Graetz I, Muelly E, Millman A, Lee C. Treatment and Follow-up Care Associated With Patient-Scheduled Primary Care Telemedicine and In-Person Visits in a Large Integrated Health System. JAMA Netw Open. 2021 Nov 1;4(11):e2132793.

2. Comment: 

Any inclusion and exclusion criteria of participants? If the "non-users" include simple patients who have good past health and only consult the clinic for episodic problems (e.g. common cold), of course they will have very few clinic visits.

Response:

We have expanded our Methods section to more explicitly address inclusion and exclusion criteria. All adult primary care patients were eligible for participation in telehealth, and we included all adult primary care patients who had scheduled visits in the study interval in our analysis. There were no set exclusion criteria for telehealth participation. Clinicians and patients used their discretion when scheduling appointments to determine whether a telehealth visit was appropriate or preferable for a given concern.

The issue of whether non-users are generally healthier patients with fewer healthcare needs (for example, coming into clinic only for an in-person “annual physical”) is a notable one. We attempted to account for this by stratifying by comorbidity count, a metric that we acknowledge is flawed for defining multimorbidity, but in the absence of disease severity data, we used this as a rough proxy. The number of comorbidities someone has in their medical record may estimate both expected healthcare utilization (i.e. more comorbidities may mean more medical service needs) and historical utilization (i.e. someone who has more engagement with the health system may accrue more diagnosis codes in their chart). We have added text and a citation in the Limitations to discuss this.

Change:

In Methods, paragraph 1: 

All adult primary care patients were eligible for participation in telehealth, and we included all adult primary care patients who had scheduled visits in the study interval in our analysis. There were no set exclusion criteria for telehealth participation. Clinicians and patients used their discretion when scheduling primary care appointments to determine whether a telehealth visit was appropriate or preferable for a given concern.

In Limitations: 

We used comorbidity count as a rough proxy for both expected healthcare utilization (more chronic diseases may correlate with more medical service use [11]) and past health system engagement (more encounters with the healthcare system may correlate with more diagnosis codes entered into a patient’s chart) and stratified by comorbidity count to account for these factors.

In references:

11. Fortuna D, Berti E, Moro ML. Multimorbidity epidemiology and health care utilization through combined healthcare administrative databases. Epidemiol Prev. 2021 Jan-Apr;45(1-2):62-71.

3. Comment: 

In the tables, I suggest to replace "something else" by "others"

Response:

We have changed “Something Else” in Table 1 to “Other” as suggested.

Change:

In Table 1, column 1: changed “Something Else” to “Other”

4. Comment: 

I think the main weakness is that there are many unmeasured parameters to explain the results. The authors may want to discuss possible important parameters and address/discuss this in more details in the discussion; e.g.

- disease severity: people with poorly controlled diseases may want more contact time with the healthcare system. Therefore, they may have more telehealth and also more consultations. No of co-morbidities is not exactly the same as disease severity

- Education level

- level of social support

Response: 

Thank you for this comment. We have added additional sentences in the Limitations regarding unmeasured confounders such as disease severity, education level, socioeconomic status, and distance to a clinic. Unfortunately, these data were not available for this study.

Change:

In Limitations:

However, telehealth users and non-users may differ in propensity to seek care for reasons unaccounted for by their number of comorbidities or the demographics measured in this study, such as distance to the clinic, education level, or socioeconomic status [12]. Further, we did not account for disease severity, which may also impact the number of encounters.

In references:

12. Reed ME, Huang J, Graetz I, Lee C, Muelly E, Kennedy C, et al. Patient Characteristics Associated With Choosing a Telemedicine Visit vs Office Visit With the Same Primary Care Clinicians. JAMA Netw Open. 2020 Jun 1;3(6):e205873.

5. Comment: 

As this is a cross-sectional study, the causal relationship cannot be established. it is possible that those patients, who had frequent clinical visits, changed some of the visits to telehealth during COVID pandemic

Response:

We have added an acknowledgement of the limitations of an observational study design in inferring causality in the Limitations section. We also added a sentence expanding on the point regarding high utilizers switching what would have been in-person visits to telehealth visits in the context of a pandemic. In our response to Comment #4, we addressed the issue of unmeasured factors that may drive that high utilization. Finally, we replaced language that may have implied causality with more associative language. 

Change:

In Abstract:

We assessed whether usage of telehealth was associated with differential primary care utilization at a large, urban public healthcare system.

In Discussion:

This suggests that availability of telehealth visits in primary care may be associated with increased individual primary care utilization.

In Limitations:

Our study examines utilization behavior during a pandemic, which may not be applicable to non-pandemic times. It is possible that some patients who were higher utilizers of the health system elected to change some visits to telehealth due to concern about travelling during a pandemic. … We used an observational study design, which limits our ability to infer causality.

In Conclusions:

Availability of telehealth services may be associated with increased primary care utilization in safety-net clinics.

6. Comment: 

The authors may also want to elaborate more about clinical/research implications of their results. RCTs and cohorts maybe a better choice to answer whether implementation of telehealth systems would increase healthcare utilization

Response:

We have added text to elaborate on the implications of our findings and areas for future research. We agree that RCTs and cohorts may be more robust ways to study associations between telehealth availability and healthcare utilization and mention this as well. 

Change:

In Discussion:

While payors and policymakers may be concerned that telehealth availability may increase healthcare costs via additive service utilization, increased primary care utilization in the context of telehealth availability may not be a negative consequence if patients can have more health concerns addressed in a safe and effective manner. Changes in reimbursement structure in the United States towards more value- or outcomes-based payment may allow health systems and other entities to provide telehealth services while tempering costs for payors regardless of visit count. 

Future studies of telehealth and healthcare utilization may consider using prospective or cohort designs to examine associations more robustly. And, further research will be needed on clinical and experiential outcomes of telehealth encounters to inform longer-term policies around provision and reimbursement of telehealth.

---

## [Decision Letter · Decision Letter 1]

25 Jul 2022

Primary care utilization among telehealth users and non-users at a large urban public healthcare system

PONE-D-22-05119R1

Dear Dr. Chen,

We’re pleased to inform you that your manuscript has been judged scientifically suitable for publication and will be formally accepted for publication once it meets all outstanding technical requirements.

Kind regards,

Esther Cubo Delgado

Academic Editor

PLOS ONE

Additional Editor Comments (optional):

Reviewers' comments:

Reviewer's Responses to Questions

**Comments to the Author**

1. If the authors have adequately addressed your comments raised in a previous round of review and you feel that this manuscript is now acceptable for publication, you may indicate that here to bypass the “Comments to the Author” section, enter your conflict of interest statement in the “Confidential to Editor” section, and submit your "Accept" recommendation.

Reviewer #1: All comments have been addressed

2. Is the manuscript technically sound, and do the data support the conclusions?

Reviewer #1: Yes

3. Has the statistical analysis been performed appropriately and rigorously? 

Reviewer #1: Yes

4. Have the authors made all data underlying the findings in their manuscript fully available?

Reviewer #1: Yes

5. Is the manuscript presented in an intelligible fashion and written in standard English?

Reviewer #1: Yes

6. Review Comments to the Author

Reviewer #1: all comments have been adequately addressed. The manuscript has improved and can be accepted for publication. thank you.

7. PLOS authors have the option to publish the peer review history of their article (what does this mean?). If published, this will include your full peer review and any attached files.

Reviewer #1: No

---

## [Editor Report · Acceptance letter]

28 Jul 2022

PONE-D-22-05119R1 

Primary care utilization among telehealth users and non-users at a large urban public healthcare system 

Dear Dr. Chen:

I'm pleased to inform you that your manuscript has been deemed suitable for publication in PLOS ONE. Congratulations! Your manuscript is now with our production department. 

Kind regards, 

on behalf of

Dr. Esther Cubo Delgado 

Academic Editor

PLOS ONE